# Research on Root Strain Response Characteristics of Inner Ring of Planetary Gear Transmission System with Crack Fault

**DOI:** 10.3390/s23010253

**Published:** 2022-12-26

**Authors:** Lan Chen, Xiangfeng Zhang, Lizhong Wang

**Affiliations:** College of Mechanical Engineering, Xinjiang University, Urumqi 830047, China

**Keywords:** planetary gear transmission system, tooth crack, fast spectral kurtosis method

## Abstract

This paper established the system dynamics model for two kinds of tooth cracks of different depths of the sun gear and inner gear ring to study the influence mechanism of crack failure on the tooth root strain of the planetary gear transmission system. Combined with the finite element model of the inner gear ring, the tooth root strain of the ring was solved. Experiments verified the correctness of the solution method. The root strain under the crack fault of the sun gear and the tooth crack fault of the inner gear ring is analyzed, and the following conclusions are drawn: Periodic fault impact occurs in the strain signal of the tooth root of the inner gear ring during the crack fault of the sun gear root. The fault can be extracted by the fast spectral kurtosis method (FSK), and the fault components are used to determine whether the sun gear cracks. The Lempel–Ziv index showed a tendency to increase gradually during the process of solar wheel crack deepening, which could be used as the damage index of crack depth. The results can provide a basis and reference for fault diagnosis.

## 1. Introduction

Planetary transmissions are widely used in aerospace, automotive, and heavy industrial applications such as helicopters, wind turbines, and heavy trucks due to their large transmission ratio and strong load-bearing capacity [1,2]. However, the planetary gear transmission system, complicated structure, and poor working conditions led to an increased equipment failure rate. The gear crack fault is an early failure, occupying a great proportion of the fault, and the internal tooth circle is an essential part of the planetary gear transmission system. Therefore, it is vital for fault diagnosis to study the influence of cracks on the root strain of the inner ring and to judge the degree of crack damage.

There have been many studies on the crack failure of the planetary gear system. In the fault simulation analysis, Chaari et al. [3] simulated the tooth pitting and cracks of the solar gear of the planetary gearbox. They analyzed the influence of the fault on the gear mesh stiffness. In addition, they compared the dynamic response of a healthy planetary gearbox with that of a planetary gearbox with eccentricity and contour errors by modeling [4]. Park et al. [5] used the finite element model to study the influence of defects on the carrier of the planetary gear set from the aspect of stress distribution. Yuksel and Kahraman [6] established a computational model to study the effect of surface wear on the dynamic behavior of planetary gearboxes. Zhang Jun et al. Han [7] analyzed the influence of ring gear crack fault on dynamic load distribution between external meshing gear pairs. Yang [8] considered the influence of tooth crack opening state on meshing stiffness and dynamic response of spur gear pair and proved that ring crack faults rarely affect the dynamic load distribution between external meshing gear pairs. Li Zhanwei et al. [9] Li Zhanwei et al. discussed the effects of the depth, length, and height of crack. The equivalent stress, contact pressure, and displacement of the tooth are also analyzed under different crack types using the FE method. Barszcz et al. [10] present the application of the spectral kurtosis technique for detecting a tooth crack in the planetary gear of a wind turbine. The authors propose a method based on spectral kurtosis, which yields good results. Their method was able to detect the existence of the tooth crack several weeks before the gear failure. Vicuna [11] proposed a phenomenological model to simulate the vibration that could be measured by a hypothetical sensor mounted outside the ring gear. Feng and Zuo [12,13] studied the spectral structure of planetary gearboxes’ vibration signals, provided a gear damage signal model, and demonstrated the signal model using experimental and industrial signals. Wu et al. [14] studied a planetary gearbox using a multi-body dynamics model. They observed that the dynamic response of the model depends on the interaction of many components inside the gearbox. Liang et al. [15] used a modified cantilever beam model to represent the external gear tooth and derive the analytical equations of the bending, shear, and axial compressive stiffness. A crack propagation model is developed, and the mesh stiffness reduction is quantified when a crack occurs in the sun or planet gear. Lei Yaguo et al. [16] studied the solution of time-varying meshing stiffness under the fault condition of a planetary wheel, established the fault dynamics model, and analyzed the spectrum characteristics under the fault condition. Xiao Zhengming et al. [17] adopted the improved energy method to establish a system dynamics model considering time-varying parameters, verified that the higher the model’s accuracy, the more pronounced the fault characteristics, and revealed the fault mechanism of early tooth root cracks. The modal sensitivity analysis is conducted using a three-dimensional dynamic model of a planetary geared rotor system for the number of planet gears, planet mistuning, mass of planet gears, gear mesh stiffness and planet gear speed [18]. Sanchez-Espiga et al. [19] proposes a numerical approach to the problem of the calculation of the load sharing in planetary transmissions by measuring the strains in the root of the sun gear teeth.

To sum up, the existing research mainly focused on various errors to reduce the influence of the whole model and the characteristic of the system of fault signal transmission. However, the study of the inner gear ring is insufficient. The planetary gear transmission system’s failure can often collect relevant fault signals on the internal gear ring. The ring gear is a critical component in planetary transmission. An easy-to-crack fault occurs. However, when analyzing fault response characteristics of vibration signals, it is easy to be affected by the transmission path. The fault components are weakened or disappear by modulation, which seriously affects the crack fault analysis. Therefore, different fault carriers are selected to reflect the fault state. As the signal carrier of fault, the strain signal is more direct than the vibration signal. The reason is that the change of time-varying meshing stiffness under crack failure will indirectly lead to the change of inner ring root strain. In contrast, the time-varying meshing stiffness caused by crack failure will directly change the strain signal of the inner ring root, which is more evident than the fault information transmitted by the vibration signal. Therefore, studying the root strain analysis of planetary transmission gear ring with the crack fault in fault diagnosis is essential.

In this paper, the dynamic model of the planetary gear transmission system was established, the failure excitation of the inner gear ring under the crack failure of the sun wheel was analyzed, the finite element model of the inner gear ring was established, the root strain of inner gear ring was solved, and the influence mechanism of root crack failure of a sun wheel and inner gear ring on the root strain of inner gear ring of planetary gear transmission was studied.

## 2. Construction of Planetary Transmission System Model

### 2.1. The Parameters of the Planetary Transmission Gearbox

This paper selects the planetary gear transmission system in the wind power generation testbed produced by SQI company for research. The straight-tooth planetary gear train is 2K-H type. The structural diagram is shown in the figure, including the sun gear S, the planet gear P, the planet carrier C, and the inner ring gear R. During the operation, the sun gear is used as the input and the planet carrier as the output components, and the four planet gears are uniformly distributed around the sun gear. The parameters of the above components are shown in Table 1. The ring gear rim thickness is 20 mm. In Figure 1, the planetary wheel is the input, and the sun wheel is the output. The gear ratio is 1.24. The planetary gear system plays an increasing role in the wind turbine gearbox.

### 2.2. Construction of System Dynamics Model

The complex coupling relationship between the planetary gear transmission system components makes modeling more challenging. Therefore, the gear transmission system is reasonably simplified. First of all, the appliance sometimes has variable stiffness of the spring links the parts, through which to reflect the meshing relationship between the parts. Secondly, the parts are regarded as rigid, and the support of the planet wheel and sun wheel and the fixing of the inner gear ring are simulated by springs with appropriate stiffness. Finally, a translation-coupled dynamic model was established without considering assembly errors, tooth clearance, and tooth machining errors.

The center of the sun wheel in the planetary gear system is taken as the center of the coordinate system, and the 100 teeth on the inner gear ring are numbered from 1 on the positive Y axis counterclockwise. Here, as shown in the figure, 1/4 of the whole inner gear ring is numbered 25 as an indication. The three degrees of freedom in transverse, longitudinal, and torsional directions are taken into account for each part of the transmission system: *f*_s_, *d*_s_, *f*_r_, *d*_s_, *f*_c_, and dc are the transverse and longitudinal micro displacements of the solar wheel, inner gear ring, and planetary frame respectively. *f*_pi_ and *d*_pi_ are the radials and tangential micro displacements of the planetary wheel. *q*_s_, *q*_pi_, *q*_c_, and *q*_r_ are the torsional micro displacements of the sun wheel, planetary wheel, planetary frame, and inner gear ring. The 21 DOF displacement array of the system is as follows
(1)$$X=\left\{f_{s},d_{s},\theta_{s},f_{\mathrm{pi}},d_{\mathrm{pi}},\theta_{\mathrm{pi}},f_{c},d_{c},\theta_{c},f_{r},d_{r},\theta_{r}\right\}$$

*k*_r_, *k*_c_, *k*_pi_ and *k*_s_ are the supporting stiffness of the inner gear ring, planet frame, planet wheel and sun wheel, respectively. *k*_rq_, *k*_cq_, and *k*_sq_ are the torsional stiffness of the inner gear ring, planetary frame, and solar wheel, respectively, where *i* = 1, 2, 3, 4.

As shown in Figure 2, when mesh stiffness of an internal spur gear pair with ring gear rim deformation is included, the relative displacement of each component in planetary transmission can be expressed as
(2)$$\left\{\begin{array}{l} \delta_{\mathrm{spi}}=(f_{pi}-f_{s})\sin\varphi_{\mathrm{spi}}+(d_{s}-d_{\mathrm{pi}})\cos\varphi_{\mathrm{spi}}+u_{s}-u_{\mathrm{pi}}\\ \delta_{\mathrm{rpi}}=(f_{pi}-f_{r})\sin\varphi_{\mathrm{rpi}}+(d_{r}-d_{pi})\cos\varphi_{\mathrm{rpi}}+d_{r}\theta_{r}-u_{\mathrm{pi}}\\ \delta_{\mathrm{cpx}}=f_{c}-f_{pi}-(u_{c}-u_{pi})\sin\varphi_{\mathrm{rpi}}\\ \delta_{\mathrm{cpy}}=d_{c}-d_{pi}-(u_{c}-u_{pi})\cos\varphi_{\mathrm{rpi}}\\ \delta_{\mathrm{spu}}=(f_{c}-f_{pi})\sin\varphi_{i}+(d_{pi}-d_{c})\cos\varphi_{i}+u_{c}-u_{\mathrm{pi}}\\ \delta_{xpi}=f_{pi}-f_{c}+u_{c}\sin\varphi_{i}\\ \delta_{ypi}=d_{pi}-d_{c}-u_{c}\cos\varphi_{i} \end{array}\right.$$
where *δ_s_*_pi_ is the projection of the relative displacement of the solar wheel and the planetary wheel along the meshing line; *s*_pi_ = *α*_s_ − *i*, where *α*_s_ is the engagement angle of the external gear pair; *δ*_rpi_: projection of relative displacement of planetary wheel and gear ring wheel along the direction of their meshing line; *r*_pi_ = *α*_r_ + *i*, *α*_r_ is the engagement angle of the internal meshing gear pair; *δ*_cpx_, *δ*_cpy_, *δ*_cpu_ are the projection of relative displacement of the planetary shelf and planetary wheel along *f*_c_, *d*_c_, and *θ*_c_ directions, respectively; *δ*_xpi_, *δ*_ypi_ are the projection of the relative displacement of the planetary wheel and the planetary shelf along the *f*_i_ and *d*_i_ directions.

The input/output torques *T*_in_ and *T*_out_ (input is positive, the output is negative) of the sun wheel and planetary frame, the mass of the sun wheel, planetary frame, planetary wheel, and inner gear ring wheel are *m*_s_, *m*_c_, *m*_p_ and *m*_r_ respectively. The moment of inertia is *I*_c_, *I*_p_ and *I*_r,_ respectively. The differential equation of dynamics can be obtained by Newton’s Equation of motion using the lumped-mass method as follows:(3)$$\left\{\begin{array}{l} m_{s}{\ddot{f}}_{s}-\sum_{i=1}^{4}k_{\mathrm{spi}}\delta_{\mathrm{spi}}\sin\varphi_{\mathrm{spi}}+k_{\mathrm{sx}}f_{s}=0\\ m_{s}{\ddot{d}}_{s}+\sum_{i=1}^{4}k_{spi}\delta_{spi}\cos\varphi_{spi}+k_{\mathrm{sy}}d_{s}=0\\ (I_{s}/{R^{2}}_{s}){\ddot{\theta }}_{s}R_{s}+\sum_{i=1}^{4}k_{\mathrm{sp}i}\eta_{\mathrm{sp}i}+k_{s\theta }\theta_{s}R_{s}=T_{\mathrm{out}}/R_{s}\\ m_{c}{\ddot{f}}_{c}-\sum_{i=1}^{4}k_{pi}\delta_{cpi}+k_{\mathrm{cx}}f_{c}=0\\ m_{c}{\ddot{d}}_{c}+\sum_{i=1}^{4}k_{pi}\delta_{cpi}+k_{\mathrm{cy}}d_{c}=0\\ (I_{c}/{R^{2}}_{c}){\ddot{\theta }}_{c}R_{c}+\sum_{i=1}^{4}k_{pi}\delta_{\mathrm{cpu}}+k_{c\theta }\theta_{c}R_{c}=T_{\mathrm{in}}/R_{c}\\ m_{p}{\ddot{f}}_{c}-k_{\mathrm{sp}i}\delta_{\mathrm{spi}}\sin\varphi_{spi}-k_{\mathrm{rp}i}\delta_{\mathrm{rpi}}\sin\varphi_{rpi}-k_{pi}\delta_{xpi}=0\\ m_{p}{\ddot{d}}_{c}-k_{\mathrm{sp}i}\delta_{\mathrm{spi}}\cos\varphi_{spi}-k_{\mathrm{rp}i}\delta_{\mathrm{rpi}}\cos\varphi_{rpi}-k_{pi}\delta_{ypi}=0\\ (I_{p}/{R^{2}}_{c}){\ddot{\theta }}_{c}R_{c}-k_{\mathrm{sp}i}\delta_{\mathrm{spi}}-k_{\mathrm{rp}i}\delta_{\mathrm{rpi}}=0\\ m_{r}{\ddot{f}}_{r}-\sum_{i=1}^{4}k_{\mathrm{rp}i}\eta_{\mathrm{rp}i}\sin(\alpha_{\mathrm{rp}i}+\varphi_{pi})+k_{r}f_{r}=0\\ m_{r}{\ddot{d}}_{r}-\sum_{i=1}^{4}k_{\mathrm{rp}i}\eta_{\mathrm{rp}i}\cos(\alpha_{\mathrm{rp}i}+\varphi_{pi})+k_{r}d_{r}=0\\ (I_{r}/{R^{2}}_{r}){\ddot{\theta }}_{r}R_{r}+\sum_{i=1}^{4}k_{\mathrm{rp}i}\eta_{\mathrm{rp}i}+k_{r\theta }\theta_{r}R_{r}=0 \end{array}\right.$$

### 2.3. Solution of Time-Varying Meshing Stiffness

ANSYS software was used to solve the meshing stiffness of the planetary gear transmission system by using the finite element method, as shown in the Figure 3. Taking the three-dimensional contact model of the planetary wheel and the solar wheel as an example, torque was applied to the driving wheel to fix the slave wheel. The meshes of the existing model are divided, and the meshes of the non-contact area are sparse. The contact surface is CONTA174, the target surface is discretized by the TARG170 element, and the main gear element is the SOLID185 element.

When the boundary conditions are applied, the total constraint is applied to the inner ring of the driving wheel, the lateral freedom of the inner ring of the driving wheel is constrained, and the torque is applied to the driving wheel. It should be noted that when torque is applied to the driving wheel, it should be applied to the position of the inner ring node of the driving wheel. The coordinate system needs to be converted into a cylindrical coordinate system, and the torque is decomposed into each node for application. In the load application process, the torque *T*_a_ to the inner ring node of the driving wheel is adopted.
(4)$$T_{a}=\frac{F_{n}r_{b}}{nr_{n}}$$
where *T*_a_ is the torque applied to a single node; *n* is the number of nodes. *r*_n_ inner ring radius of the main driving wheel; *r*_b_ is the radius of the base circle of the main driving wheel.

The inner ring deformation *μ*_1_ of the driving wheel was calculated by the deformation program, the driving wheel deformation Angle *δ*_1_ was obtained by the formula, and the equivalent deformation *μ* of the meshing line was finally obtained.
(5)$$\left\{\begin{array}{l} \sigma_{1}=\frac{\mu_{1}}{r_{n}}\\ \mu =\delta_{1}r_{b} \end{array}\right.$$
where *δ*_1_ is the main wheel deformation angle, *r*_n_ is the main wheel inner ring radius, *μ*_1_ is the main wheel inner ring deformation, *μ* is the equivalent deformation at the meshing line.

Meshing stiffness is calculated as follows
(6)$$k_{n}=\frac{F_{n}}{\mu }=\frac{T}{\theta r_{b}^{2}}$$

The meshing period n of a tooth on the sun wheel is divided equally, and the meshing angle of the planet wheel is adjusted according to the meshing period after the equal division to obtain the meshing state of n sun wheels and the planet wheel. In addition, the meshing period n of each tooth on the planetary wheel was equally divided, the meshing angle of the planetary wheel was adjusted according to the meshing period, and the meshing state between the planetary wheel and the inner gear ring was obtained.

The solution result through the meshing stiffness calculation formula is shown in Figure 4 and Figure 5. As can be seen from the figure, the meshing process presents an alternation of two-tooth meshing to single-tooth meshing to two-tooth meshing. In the alternation process, meshing stiffness decreases sharply when meshing to a single tooth, and stiffness excitation is extremely significant.

### 2.4. Calculation of Gear Fault Stiffness

During the planetary gear transmission system operation, the gears are constantly engaged, and the gear teeth receive the continuous impact of the meshing excitation. Structural fatigue and cracks easily occur at the gear’s root, as shown in Figure 6, e is the crack depth, and *g*_c_ is the crack angle. The meshing stiffness calculation method in Section 2.3 is used to establish a gear model with a tooth root crack here to solve the time-varying meshing stiffness under the fault of the tooth root crack. In the past, Refs. [9,10] used different gear crack modeling methods to calculate the stiffness of cracked gear. In this paper, the crack modeling method was simplified to facilitate the modeling and solving, and the crack model with uniform propagation depth along the tooth width was established. When cracks appear, the gear’s moment of inertia and effective cross-sectional area change, and the gear tooth model has greater flexibility. Therefore, the time-varying mesh stiffness decreases with the increase in crack depth. The meshing stiffness of the gear with the crack fault is shown in Figure 7.

## 3. Dynamic Load Analysis of Crack Fault

The solar wheel was used as the input, the speed was set at 600 r/min, and the load was 500 N·m. The meshing excitation was solved and analyzed. The Figure 8 shows that the dynamic excitation under normal working conditions shows periodic fluctuations. The dynamic excitation fluctuates up and down around the mean value of 2145 N, with a fluctuation period T of 0.00457 s and a fluctuation amplitude of 25 N. When there is a solar wheel crack fault, periodic fault impact appears in the dynamic excitation time domain, and the impact characteristics are instantaneous decrease and instantaneous increase. The instantaneous reduction of impact characteristics is caused by the instantaneous reduction of meshing stiffness when the fault tooth engages with a planetary wheel. Meanwhile, the dynamic excitation decreases with the decrease in stiffness, as shown in Figure 9b. At the same time, the dynamic excitation of the other three planetary wheels will increase instantaneously. This increase is relatively small compared to reducing the planetary wheels’ dynamic excitation with fault teeth engagement. According to the time domain analysis, the dynamic excitation still fluctuates around the mean value of 2145.61 N in the fault condition, and the time interval between the fault tooth engaging the same planetary wheel again is a complete fault cycle *T*_g_ = 0.128 s. At the same time, with the increase of crack depth, the mean value of dynamic excitation remains unchanged, and the fault impact amplitude increases.

Solar wheel single tooth crack fault, dynamic excitation fault frequency is solved as follows
(7)$$f_{\mathrm{gs}}=(n_{s}-n_{c})/60$$
where *n*_s_ is the sun wheel speed, *n*_c_ is the rotational speed of the planetary frame. In the time domain history of dynamic excitation, every time the solar wheel meshed with the same planetary wheel, the meshing excitation suddenly decreased once and increased three times. In the corresponding frequency domain analysis, three kinds of fault signals, *f*_gs_, 2*f*_gs_ and 3*f*_gs_, appeared. In the operating condition of 600 r/min speed, *f*_gs_ was 7.81 HZ, and frequency coupling appeared at 4*f*_gs_ with a relatively small amplitude. Crack faults lead to fault components in the low-frequency band of the signal spectrum, and the amplitude of fault components increases with the increase of crack depth.

## 4. Solution of Response of Finite Element Model of Inner Ring

The tooth root crack of the sun gear and the tooth root crack of the inner ring was selected to analyze, aiming at the crack failure of the planetary gear transmission system. Analysis to solve the sun round the inner ring gear vibration response under the crack fault, the finite element model of trouble-free with the internal ring gear, the sun wheel crack fault conditions of extracting dynamic incentives, according to the tooth-bearing contact analysis (LTCA) uniform, applied to tooth surface contact, by ordinal position in tooth surface applied load equivalent gear meshing process. The dynamic equation of the inner ring under the action of meshing excitation is:(8)$$M\ddot{v}+c\dot{v}+Kv=G$$
where *M* is the mass matrix, *C* is the damping matrix, *K* is the stiffness matrix, *V* is the displacement vector of the node, *G* is the external load matrix.

For modal extraction of the finite element model, the first 50 modal matrices were selected for decoupling, and then the Newmark-β time integral method was adopted to solve Equation (8).

The main location of the solar wheel crack fault is the tooth of the solar wheel, so the dynamic meshing excitation under the fault condition is collected and applied to the finite element model of the inner tooth ring to solve the fault vibration response, as shown in Figure 10. The fault location of the inner gear ring crack is on the inner gear ring, and the dynamic meshing excitation under normal working conditions is collected and applied to the finite element model of the inner gear ring crack to solve the problem.

## 5. Experimental Verification

The wind turbine test bench produced by SQI company was used for the experiment, which was composed of a magnetic powder brake, two-stage fixed-axis reducer, wind power planetary accelerator, and drive motor. Strain signals were collected by a grating dynamic monitoring system, as shown in Figure 11.

In order to analyze the accuracy of the simulation signal, collect the time-domain history of the experimental strain signal and compare it with the simulation signal, observe the period of the time-domain history of the strain and the change of the strain value, compare the error between the simulation signal and the experimental signal, set the speed of the input shaft of the sun gear at 52.2 r/min and the load torque at 59 N·m for simulation and experiment, measure the strain data in the experiment respectively and simulate and solve the root strain signal. The time domain history comparison results of tooth root strain are shown in the Figure 12.

Comparing the simulation signal with the experimental signal, the time domain history of the signal can be divided into the meshing region and the non-meshing region. Meshes into the recess action of the meshing area of the planet wheel sensor location, rim tooth root, first squash and then stretch the process. As the chart shows, the simulation signal and experimental signal strain signals are in compression and tension of the trend, but as a result of the experiment are certain errors in the test method, the simulation results of the maximum tensile strain for 29 με. In the experiment, the maximum tensile strain is 26 με, and the error is 10.3%. In the non-meshing region, the experimental signal is compared with the simulation signal, and the simulation signal decreases to near zero with a slight fluctuation. In the experimental signal, the strain attenuation is relatively slow because the gear ring gradually recovers to the state before deformation when the gear teeth are gnawing out. The error is within tolerable limits. Therefore, the experimental results show that the simulation results are reliable.

## 6. Strain Analysis

### 6.1. Analysis of Failure Strain Results of Sun Gear Tooth

The dynamic strain method was used to solve the crack fault of the planetary gear transmission system. The crack fault of the solar wheel affects the strain signal results of the root of the inner ring of the planetary transmission system, as shown in the Figure 13.

Solving process, as shown by the sun around a crack of the gear fault, the fault gear and planetary gear, the meshing stiffness of volatility and fault gear meshing planetary wheel at the same time with the inner ring gear meshing, affected by the fault gear, planetary wheel and the ring gear meshing to stimulate a cyclical fluctuation, fluctuation cycle as shown in Figure 8. The input speed was set at 600 r/min and the load at 500 N·m. According to the finite element model solution process, the fault meshing excitation was applied to the finite element model of the inner gear ring. The solution results of strain extraction were as following Figure 14.

Under normal working conditions, because the planetary gear transmission system is a structure of four planetary wheels, and the four planetary wheels keep rotating with the planetary frame, the planetary wheels engage the tooth root strain collection position one by one. The strain shows periodic fluctuation, with the fluctuation cycle *T*_r_ being 0.1142 s, namely a quarter of the mesh cycle. The inner gear ring is supported by eight evenly distributed (Kr) fixed. As shown above, the tooth root stress extraction location selection between the adjacent two support and the time domain history can be divided into planetary gear extraction into the two supporting points to lead mesh deformation. The mesh over the strain extracts the location where the two supports are caused by the deformation of two kinds of vibration signal fluctuation.

By analyzing the time domain history of normal and fault working conditions, the dynamic excitation decreases when the sun gear tooth cracks occur, which further leads to the change of the tooth root strain. As shown in Figure 15, when the planetary wheel was about to engage two teeth near the extraction position of tooth root strain, it was not affected by the faulty teeth. Under the influence of compression deformation of the tooth ring, the compressive strain was not significantly different from that under normal working conditions. As the planet gear meshed through the tooth root strain extraction position, the cracked tooth meshed with the planet gear simultaneously, and the tensile strain weakened compared with the normal working condition. In dynamic excitation analysis, increased crack depth leads to different weakening degrees of dynamic excitation. Dynamic excitation weakens compared with normal working conditions, and the failure tensile strain also decreases, and the strain reduction degree is inversely proportional to the crack depth.

The root strain of the fault conditions a significant change in the mesh deformation area. This sets the planet wheel failure tooth contact with the sun at the same time and the inner gear tooth root circle strain mesh patch for the initial position until the planet wheel and inner gear ring patch and fault signal tooth meshing *T*_g_ for complete failure cycle at the same time. Again the sun wheel crack failure strain cycle for the *T*_g_ solution is as follows
(9)$$T_{g}=\frac{\left[Z_{s}+Z_{r},Z_{s}\right]}{n_{s}*N*Z_{s}}$$
where *Z*_s_ and *Z*_r_ are the number of teeth of the solar wheel and planetary wheel, *N*_s_ is the sun wheel speed, and *N* is the number of planetary wheels.

As shown in Figure 15 (enlarging the deformation by 10,000 times), in the meshing deformation zone, when the planetary wheel was about to engage the sun wheel fault tooth, the tooth teeth on both sides of the strain collection point were affected by the compression deformation of the tooth ring. A large compressive strain appeared at the extraction position of the tooth root strain. As the planetary wheel meshed through the fault tooth, the meshing excitation caused the tensile deformation of the strain collection point, and the strain signal increased sharply. After the planetary wheel gradually gnaws the fault tooth, the strain signal fluctuates around 0. Analysis of tooth root stress wave curves under different meshing states because the internal gear ring structure has symmetry. The number 1~25 teeth are the research object, the tooth root position near the alveolar strain signal acquisition and analysis of the failure of meshes into gear. The meshing tooth and tooth broke down after the tooth of the three states of strain signal, setting the fault tooth in meshing, The planetary wheel meshes with 7~8 teeth on the inner ring. As shown in Figure 16, combined with trend analysis, the variations of dynamic incentive at will, while the link meshes into the tooth state failure within the planet wheel and near six teeth on the ring gear meshing, 5 and 6~8 tooth root position before subjected to tensile deformation, the rest of the tooth root position affected by compression deformation and dynamic incentive to keep normal at this time, so the working condition of two kinds of strain signals keep highly consistent. In fault gear meshing condition, 8 tooth root stress signal under the effect of fault appeared weak. It is caused by fault under dynamic excitation attenuation compression deformation is reduced, compared to the compressive strain of collected signals under normal working conditions of a less. The rest of the tooth’s position by compressive deformation is reduced, compressive strain decreases, and the tensile deformation decreases at the same time, as well as the decrease of strain. After the failure teeth were gnawing, the dynamic excitation remained normal, and the groove position of 9~10 teeth was subjected to compression deformation. The root strain had little difference from the normal condition, and the tension strain signals of other root positions did the same. The two adjacent supports (14 teeth~25 teeth) in the three states are in the vibration deformation zone. Due to the support’s fixed action, the tooth root’s strain signal always fluctuates weakly around 0, and the deformation is small.

### 6.2. Fault Strain Analysis of Inner Ring Teeth Crack

The specific flow of strain solution for cracks at the root of the inner ring is as following Figure 17.

Compared with the crack fault of the sun wheel, the meshing excitation under the crack fault of the tooth root of the inner ring is normal. Therefore, the system dynamics model is set at the normal working condition of 600 r/min speed and 500 N·m load. The meshing excitation is extracted, and the finite element model with cracks at a tooth root of the inner ring is established. Finally, the corresponding dynamic strain method calculates and analyses the results.

According to the symmetry of the gear transmission system, the 25 teeth marked on the inner ring were taken as the research object. The crack diagram of the inner ring tooth is shown in Figure 18, and the time domain history of the tooth groove strain was collected for comparative analysis, as shown in Figure 19.

Mesh deformation zone and local amplification, as shown in Figure 20. This can be divided into four a, b, c, d, and e (five) parts. The planetary wheel meshes into fault support but does not mesh into the tooth in the strain extraction location (Figure 20a). Figure 21 state 2, namely the planet wheel mesh to signal extraction location, signal acquisition position affected by gear ring tensile deformation, and tooth root part of the strain increase. The planetary wheel is about to engage in the strain acquisition position (Figure 20b), and the gear ring is convex and deformed at this position. The strain signal shows a sharp decrease under the action of compression deformation at the tooth groove position. The overstrain acquisition position of the planet wheel meshing (Figure 20c) and the tooth groove position was subjected to tension and compression deformation simultaneously. The strain increased suddenly, corresponding to state 3 in Figure 21. The planetary wheel is about to snap out the location of signal extraction (Figure 20d), which is subjected to tensile deformation, and the strain signal decreases. After the planetary wheel gnaws out the signal collection point (Figure 20e), the strain signal fluctuates near 0 with the deformation of the inner gear ring.

In the meshing deformation area, the crack fault on the inner ring changes the original inner ring structure. Dynamic excitation is applied to the tooth surface in the planetary gear meshing failure process, and larger deformation is concentrated at the crack position. As a result, the amplitude of compression deformation and tensile deformation at the position of the crack tooth groove decreases to a certain extent. When the fault tooth is located in the vibration deformation area, corresponding to state 1 in Figure 21, the strain phase at the tooth groove also appears to have certain weaknesses in normal working conditions.

In total, 25 teeth, including fault teeth on the tooth ring, were selected to be numbered, and a strain fluctuation curve was drawn. Typical states are shown in Figure 22. When the planetary gear meshes to state 2, the fault tooth does not engage, the position of the fault tooth is affected by the compression deformation of the tooth ring, and the tensile strain decreases. When meshing to state 3, dynamic excitation is applied to the fault tooth. The meshing stiffness decreases, and the tension strain and compression strain in the meshing deformation zone decreases. When the planetary gear runs to state 4, the corresponding position of the fault tooth is affected by the tensile action of the tooth ring, and the tensile strain decreases.

The planetary gear tooth crack fault is similar to the sun gear tooth crack fault, both changing the meshing excitation between the planetary gear and the inner gear ring. When the inner gear tooth crack fault occurs, the meshing excitation is not affected, so it is classified as the planetary gear and sun gear crack fault.

### 6.3. Crack Damage Discrimination

According to the strain signal of the inner ring under crack fault, crack damage discrimination was carried out on the planetary gear transmission system, and the specific process is as following Figure 23.

According to the fault mechanism analyzed above, when the sun wheel crack fault occurs, it causes a periodic fault impact on the strain signal of the tooth root on the tooth ring. The expression of the inner tooth root crack fault is that the amplitude of the strain signal caused by the fault tooth changes. Therefore, the root strain signal is first collected, and the fault type is distinguished according to the fault frequency component in the time domain signal. If there is no obvious fault frequency in the signal component, then the crack fault on the inner ring is judged by comparing the amplitude fluctuation of the normal signal.

After the strain signal collection, the signal components are analyzed. Kurtosis is sensitive to impact signals and can be used to evaluate the strength of vibration impact components such as bearings and gears. However, the kurtosis index is easily disturbed by noise and cannot reflect the change characteristics of the impact. In order to solve this problem, the spectral kurtosis (SK) index was proposed in recent years, which can reflect the intensity of the transient impact and indicate its frequency [20]. It is a method that combines the kurtosis index with spectrum analysis, which can detect fault impact signals very effectively. This method was first introduced into the field of fault diagnosis by Dwyer [21] as a theoretical statistical method. After that, Antoni. J [22] provides the formal theoretical basis for applying SK to fault diagnosis, thus filling the theoretical basis blank of SK in the field of fault diagnosis. A spectral kurtosis method based on the short-time Fourier transform (STFT) is proposed and applied to diagnose actual mechanical faults, as shown in Figure 24.

The fast spectral kurtosis diagram is analyzed. The maximum kurtosis value is 12.6, the corresponding center frequency is 815 Hz, and the bandwidth is 328 Hz. The processed fault signals are shown in the Figure 25 below.

According to the sun gear tooth crack fault cycle, *T*_g_ solved above is the corresponding fault frequency *f*_g_. In the signal envelope diagram of the sun gear tooth crack, the fault frequency, and its frequency doubling are obvious.

After processing the fault signal of the inner gear ring crack, the main components are four times the planetary frame rotation frequency and frequency doubling, and meshing frequency and no obvious fault components appear.

The fast spectrum kurtosis processing results of the fault strain signals of the sun wheel crack and the inner gear ring crack were compared, which was consistent with the characteristics of the above analysis. When the crack fault was determined, the fault component could be identified as the sun wheel or the inner gear ring according to the characteristics.

According to the signal characteristics of different parts, different methods are used to judge the degree of crack damage. If there is a solar wheel fault impact component in the crack fault signal, the complexity index sensitive to the periodic fault impact is selected to determine the damage degree.

As shown in Figure 26, 1, 2 and 3, respectively, refer to the three states of the normal, shallow crack fault, and deep crack fault. With the increase of crack depth, the index increases accordingly. Therefore, this index is used to measure the damage degree of the sun gear tooth under crack conditions.

If it is determined that there is no fault impact component in the crack fault signal, then the signal amplitude is further compared. Under the known crack fault state of the inner ring, the tension-compression deformation trend in the strain signal decreases. With the increase of crack depth, the tension-compression strain value decreases more. According to the characteristic of the fault strain of the inner ring crack, the root mean square value, which is sensitive to signal amplitude change, is selected to judge the degree of crack damage.

Figure 27 shows states 1, 2, and 3 represent the normal, shallow, and deep inner ring cracks, respectively. As the amplitude of the strain signals in the crack state changes, the changing trend of the signal amplitude is described by the root mean square value. As the crack depth increases, the root-mean-square value of strain signals in the inner ring crack root decreases continuously. This verifies the above analysis that the strain signals in the inner ring crack depth increase continuously.

## 7. Conclusions

The system dynamics model was established by taking the crack fault in the gear ring of planetary transmission as the research object. The strain signal was solved by combining the finite element model of the inner gear ring. The correctness of the model was verified through experiments, and the crack fault was added to analyze the strain signal of the solved tooth root, and the conclusion was reached:(1)When the sun wheel tooth cracks occur, the fault signal of tooth root strain will occur only if the tooth has meshed with the fault tooth. In contrast, when the tooth root cracks occur in the inner ring tooth, the strain at the tooth root position will fluctuate due to the influence of the crack, regardless of whether there is a planetary wheel meshing with the cracked tooth.(2)The strain signals under the sun wheel and inner ring crack fault can be distinguished by the fast spectrum kurtosis method (FSK).(3)According to the strain signal characteristics of the sun wheel’s crack fault and the inner ring’s crack fault, the damage degree of the crack fault of the sun wheel can be identified by the Lempel–Ziv algorithm. The damage degree of the crack fault of the inner ring can be identified by the root mean square index.

In this paper, the finite element analysis model considering the flexibility of the gear ring is established to study the crack damage response characteristics of planetary transmission systems from the vibration signal and strain signal of two fault carriers. However, there are still other aspects to be studied. (1) This paper analyzes the planetary gear transmission crack fault. However, the effects of planetary gear transmission crack propagation in different directions on system performance and the life changes of gear components after propagation can be further studied. (2) This paper analyzes the fault response characteristics of the crack and does not conduct quantitative research and discrimination on the fault damage degree. Subsequent research can be carried out for the crack damage degree discrimination.

## Figures and Tables

**Figure 1 sensors-23-00253-f001:**
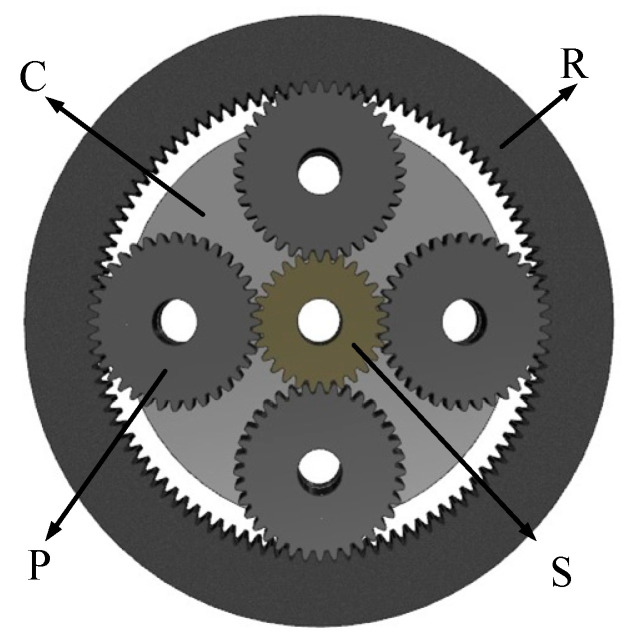
Schematic diagram of the planetary gear train structure.

**Figure 2 sensors-23-00253-f002:**
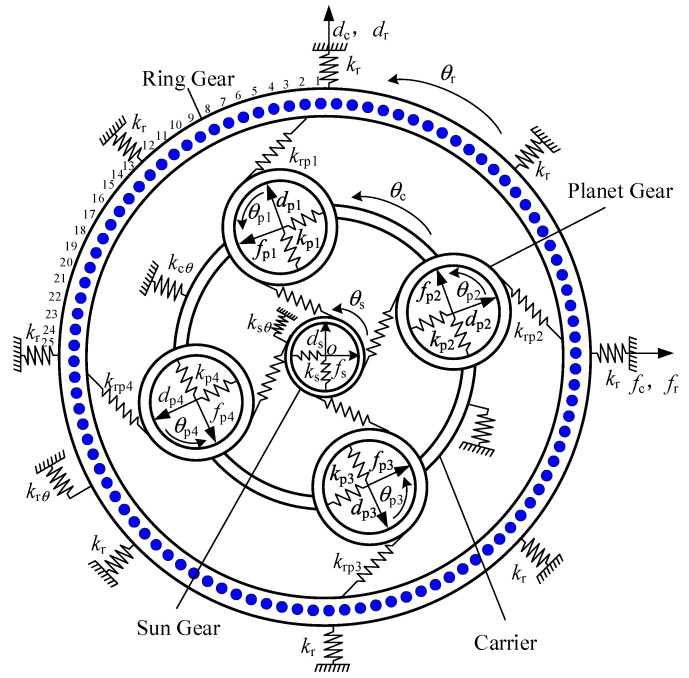
System dynamics model.

**Figure 3 sensors-23-00253-f003:**
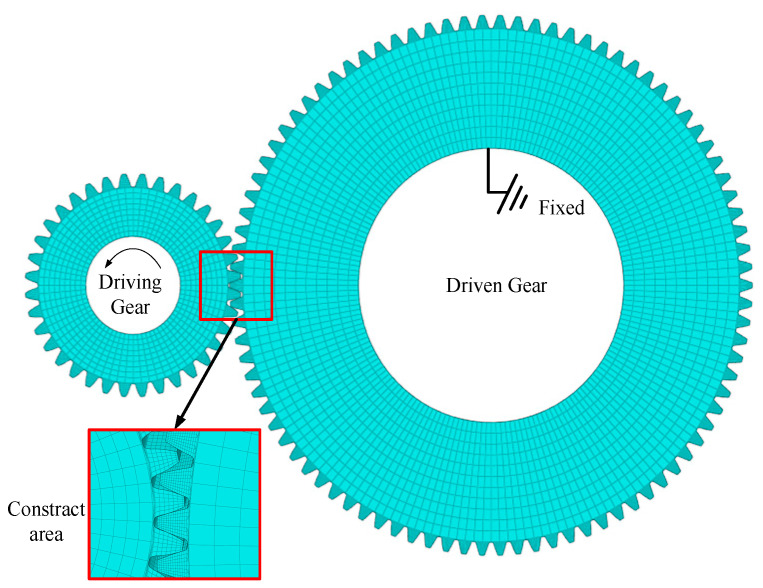
Meshing stiffness solution model.

**Figure 4 sensors-23-00253-f004:**
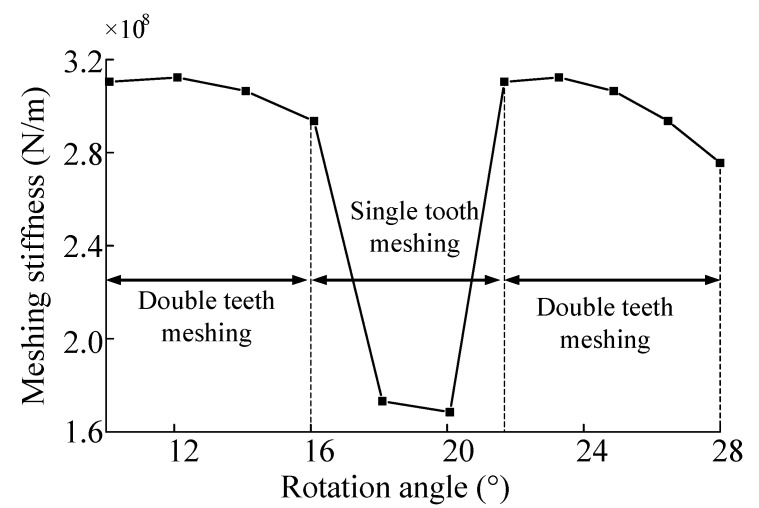
Meshing stiffness under normal conditions.

**Figure 5 sensors-23-00253-f005:**
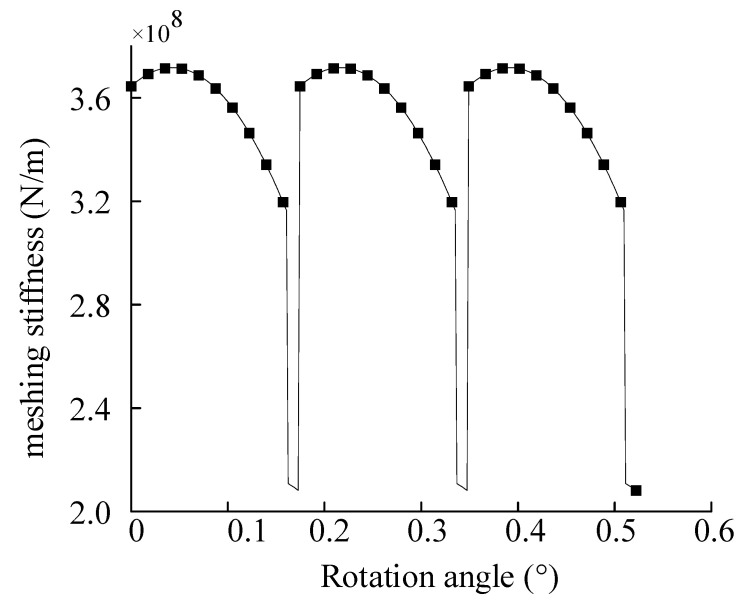
The meshing stiffness of inner gear ring and planetary gear.

**Figure 6 sensors-23-00253-f006:**
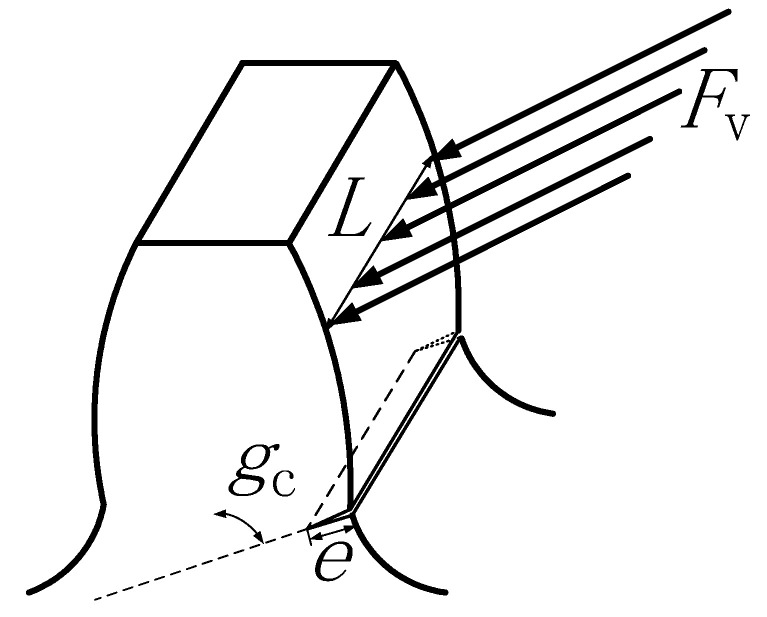
Meshing root crack model.

**Figure 7 sensors-23-00253-f007:**
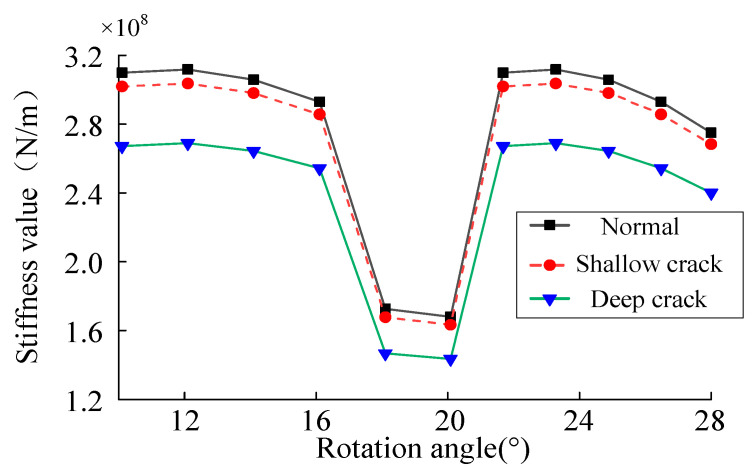
Meshing stiffness under different fault states.

**Figure 8 sensors-23-00253-f008:**
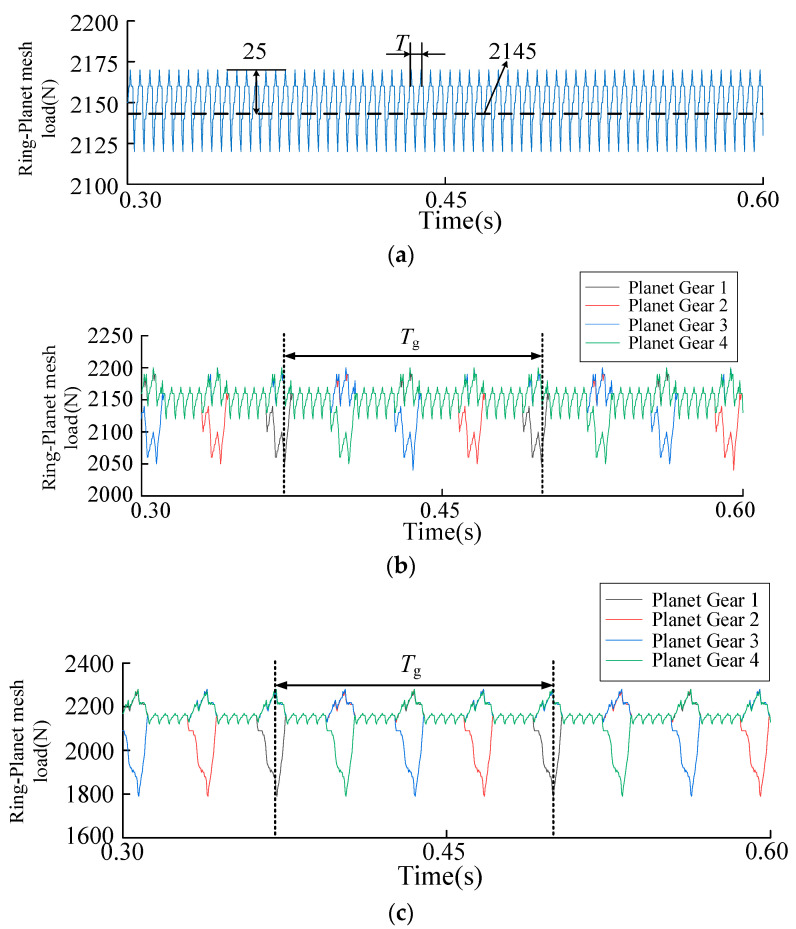
(**a**) Time domain history of the normal dynamic loads. (**b**) Time domain history of the shallow crack dynamic load. (**c**) Time domain history of the deep crack dynamic excitation.

**Figure 9 sensors-23-00253-f009:**
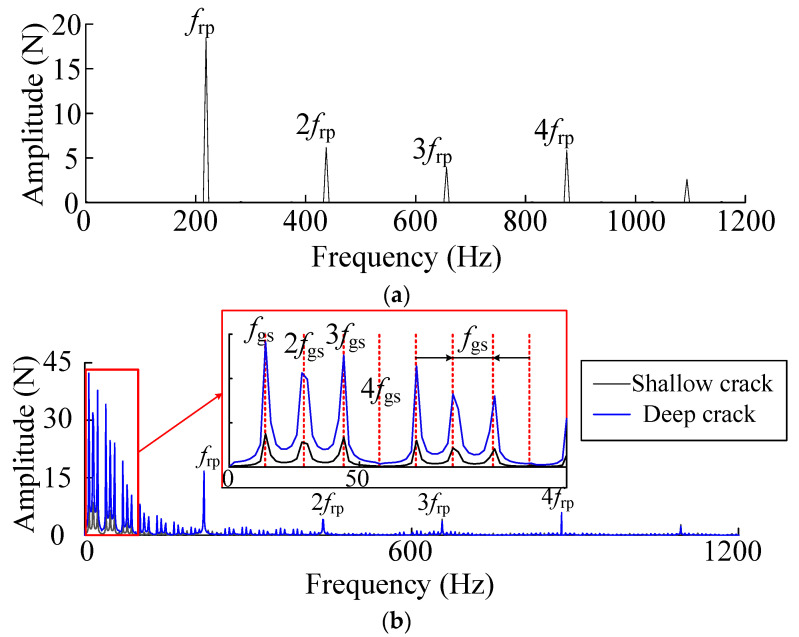
(**a**) Frequency domain history of the normal dynamic loads. (**b**) Comparison diagram of the dynamic excitation frequency domain under crack fault.

**Figure 10 sensors-23-00253-f010:**
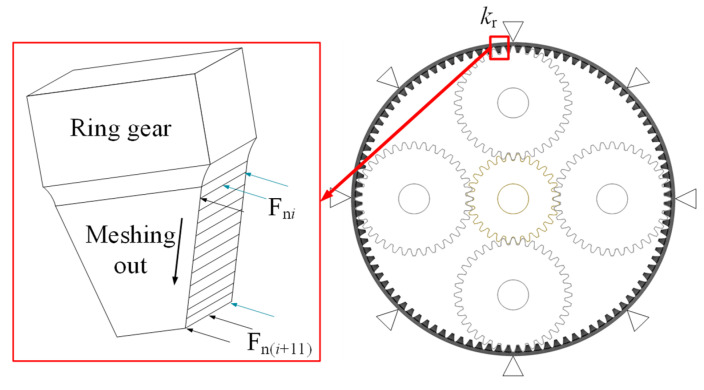
Finite element model and schematic diagram of dynamic load application.

**Figure 11 sensors-23-00253-f011:**
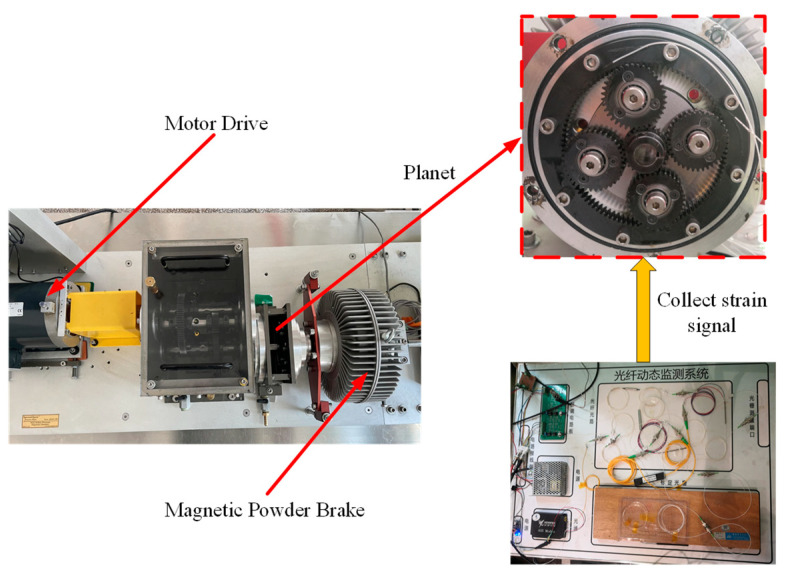
Wind turbine test bench and signal acquisition system.

**Figure 12 sensors-23-00253-f012:**
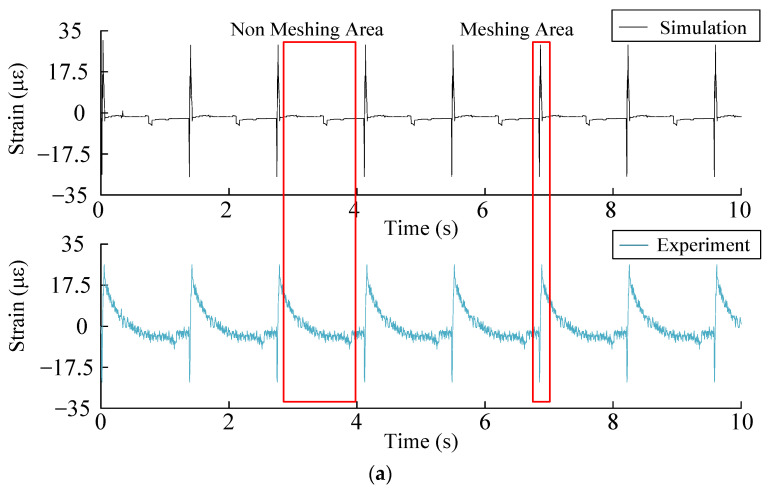
(**a**) Simulation and experiment signal time domain history. (**b**) Enlarged diagram of the meshing area.

**Figure 13 sensors-23-00253-f013:**
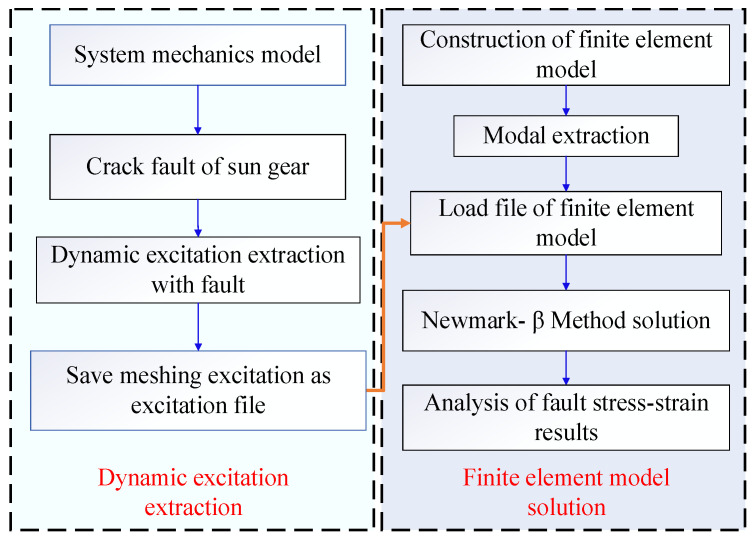
Flow chart of the dynamic strain solution of the inner ring under the crack fault of the solar wheel.

**Figure 14 sensors-23-00253-f014:**
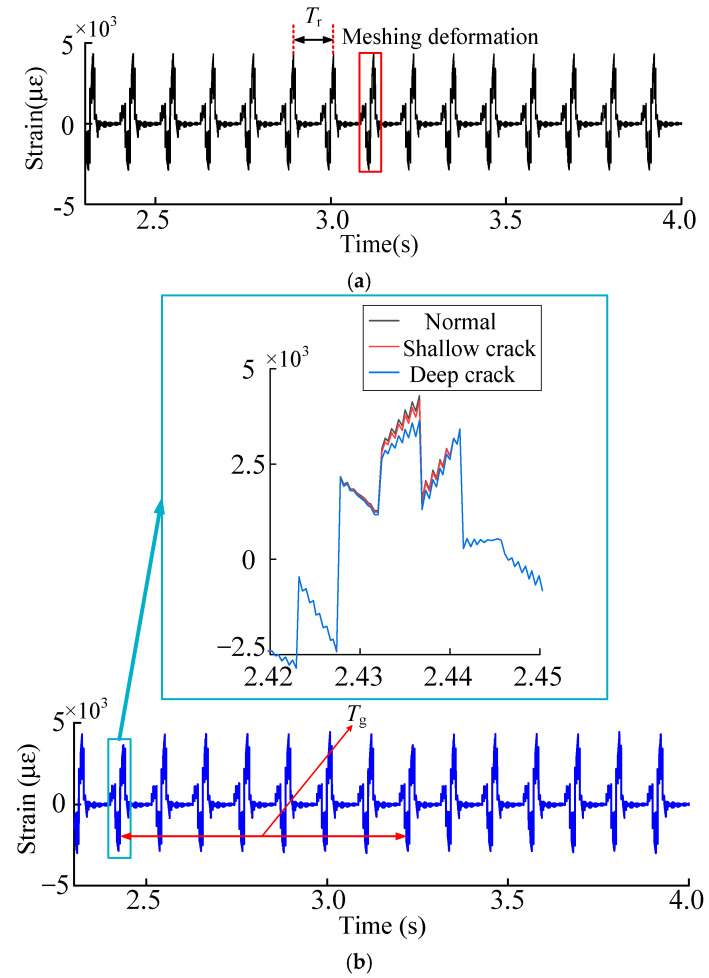
(**a**) Time domain history of the gear ring root strain under normal working conditions. (**b**) Time domain history of tooth root strain of sun gear under crack condition.

**Figure 15 sensors-23-00253-f015:**
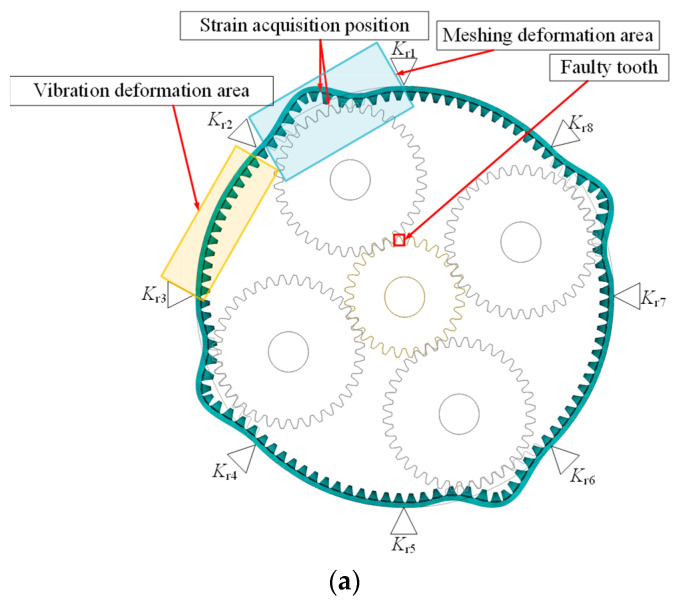
(**a**) The planetary gear is about to snap into the faulty tooth inner ring deformation. (**b**) Planetary gear meshing fault inner ring deformation. (**c**) The inner ring is deformed before the planetary gear gnawing the faulty tooth.

**Figure 16 sensors-23-00253-f016:**
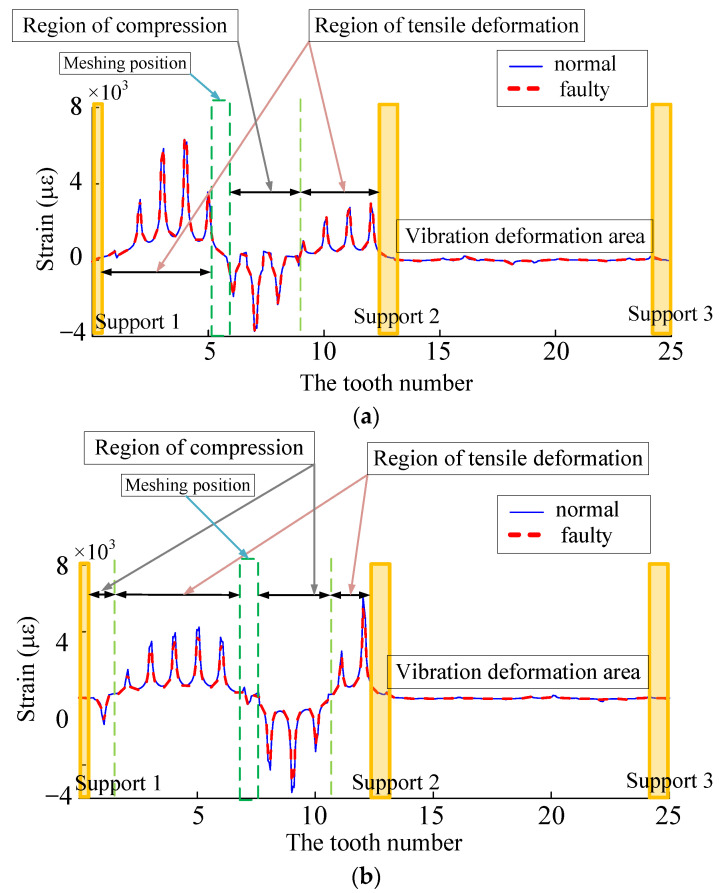
(**a**) The stress fluctuation curve of the fault tooth is about to be inserted. (**b**) Strain fluctuation curve of the snagging fault tooth. (**c**) Strain fluctuation curve of the gnawed fault tooth.

**Figure 17 sensors-23-00253-f017:**
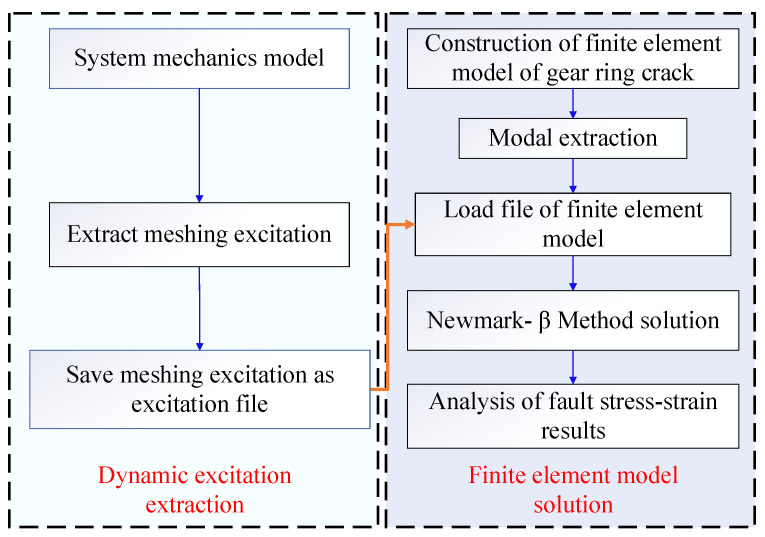
Flow chart of dynamic strain solution for inner ring root crack fault.

**Figure 18 sensors-23-00253-f018:**
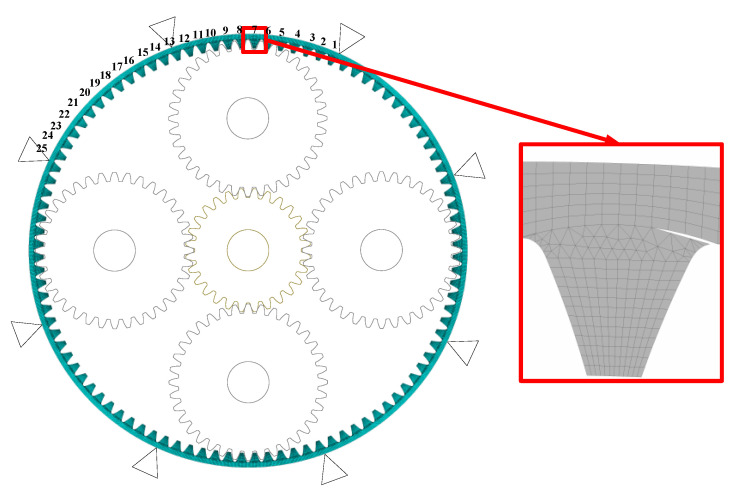
Diagram of tooth root crack model of the inner ring.

**Figure 19 sensors-23-00253-f019:**
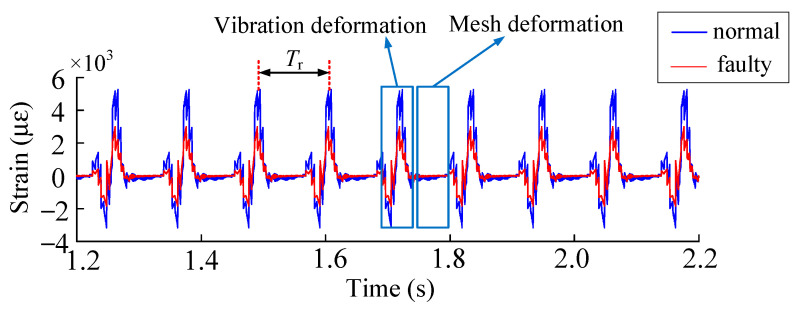
Time-domain history diagram of the strain signal of the inner ring tooth root crack.

**Figure 20 sensors-23-00253-f020:**
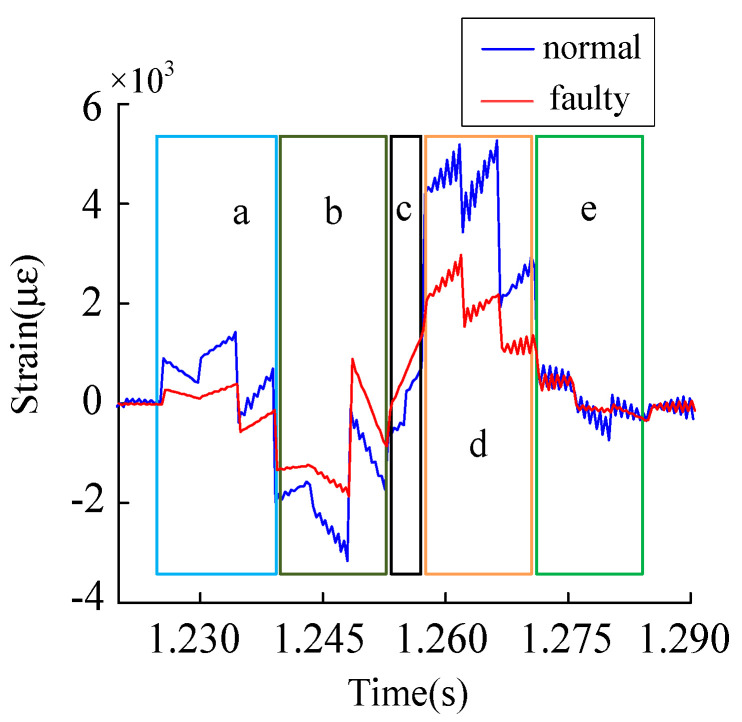
Enlarged view of the partial meshing deformation zone.

**Figure 21 sensors-23-00253-f021:**
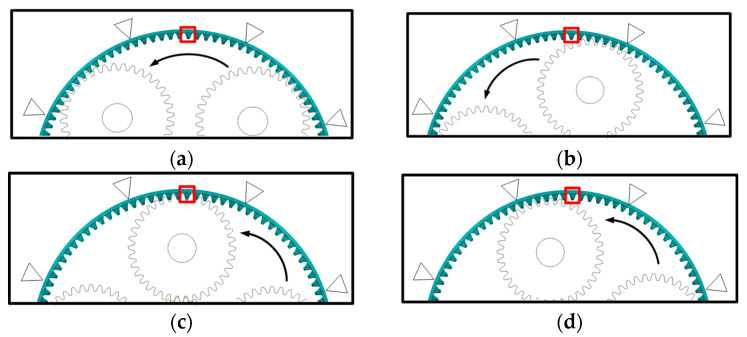
(**a**) State 1. (**b**) State 2. (**c**) State 3. (**d**) State 4.

**Figure 22 sensors-23-00253-f022:**
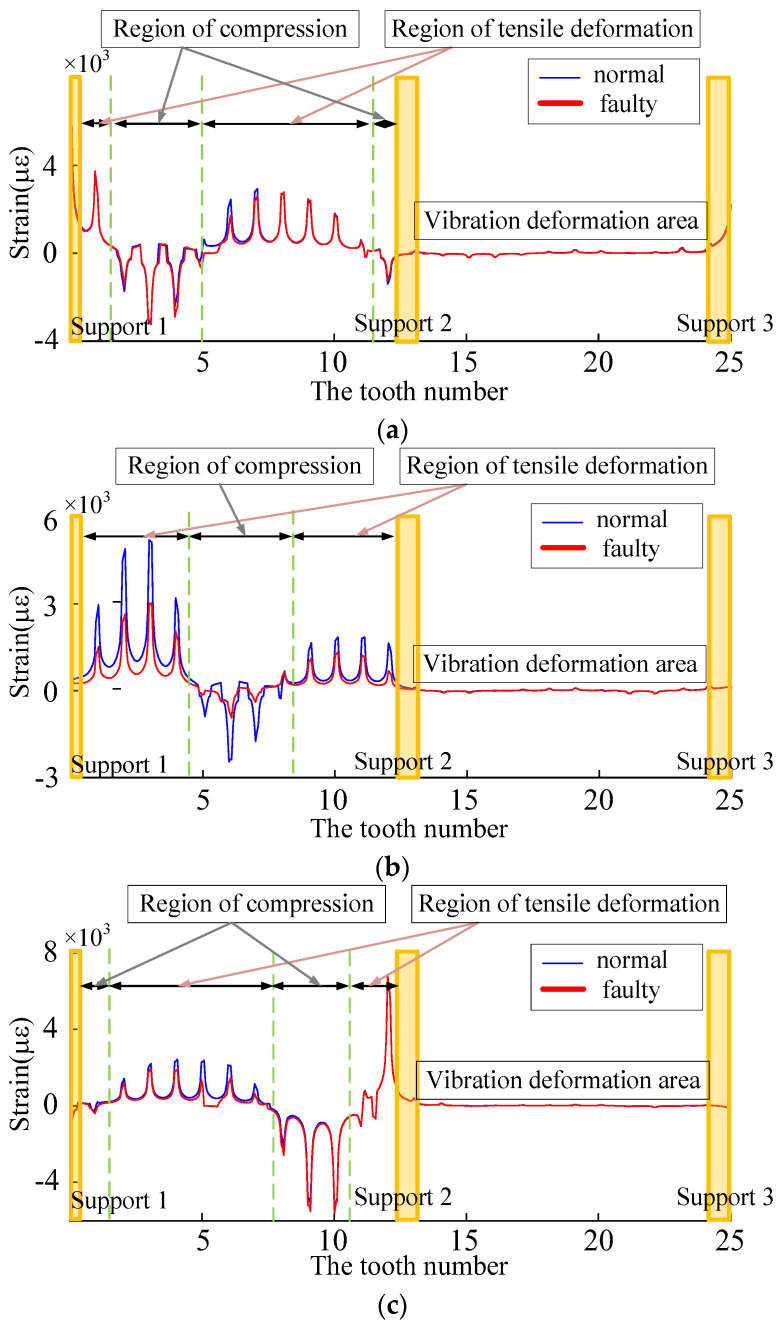
(**a**) State 2 strain wave curve. (**b**) State 3 strain wave curve. (**c**) State 4 strain wave curve.

**Figure 23 sensors-23-00253-f023:**
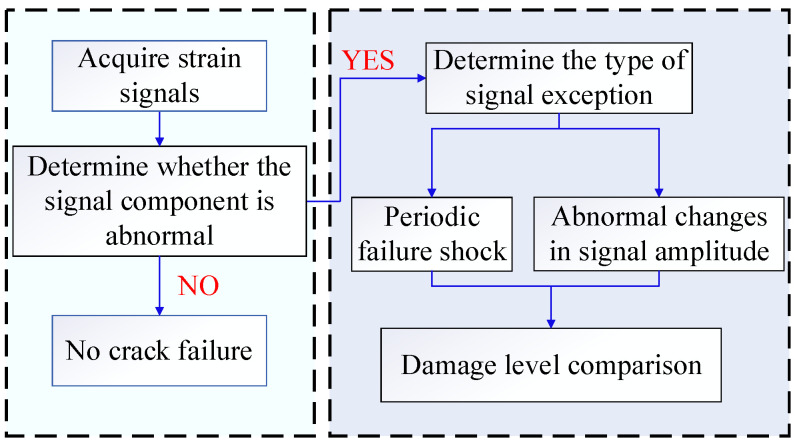
Damage discrimination flow chart.

**Figure 24 sensors-23-00253-f024:**
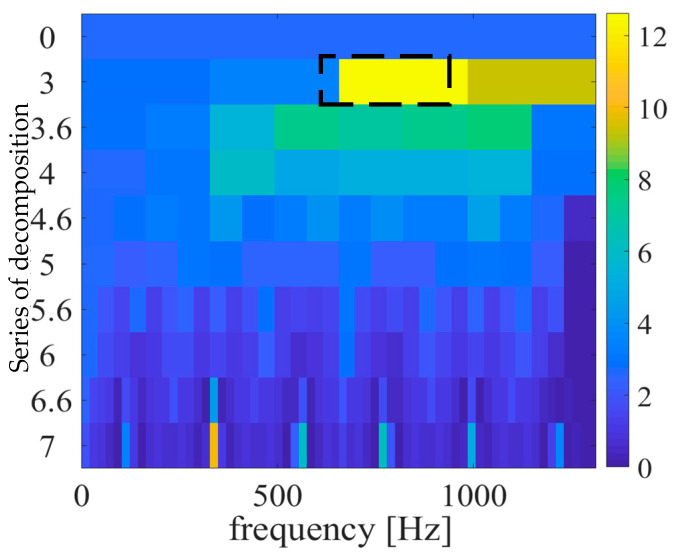
Fast spectral kurtograph.

**Figure 25 sensors-23-00253-f025:**
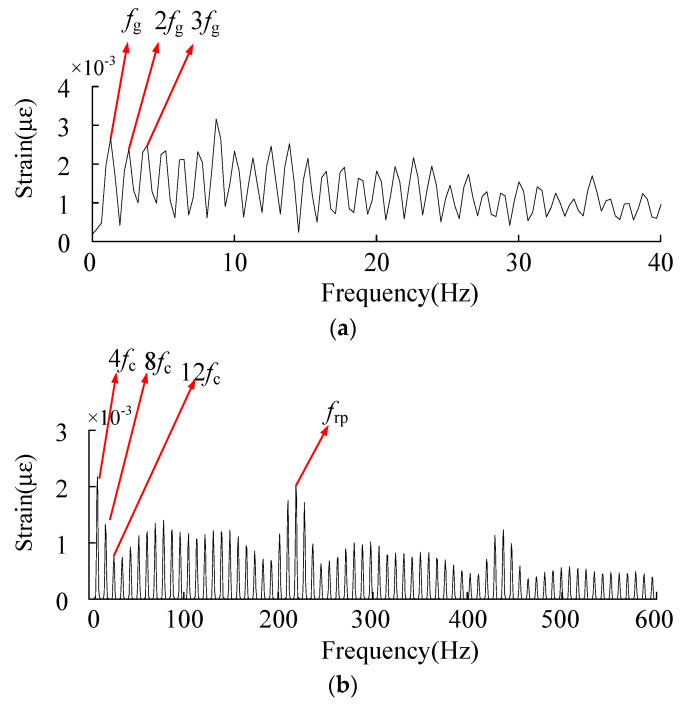
(**a**) Sun gear tooth crack fault signal. (**b**) Inner ring crack fault signal.

**Figure 26 sensors-23-00253-f026:**
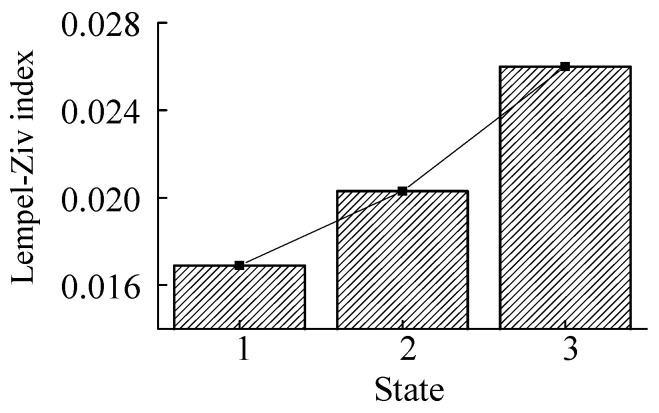
Complexity in different states.

**Figure 27 sensors-23-00253-f027:**
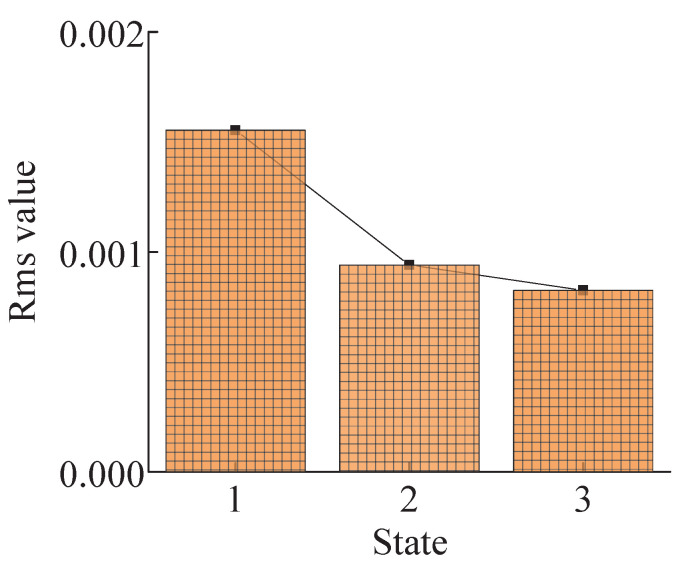
Root-mean-square values in different states.

**Table 1 sensors-23-00253-t001:** Planetary gear transmission system parameters.

Name	Sun	Planet	Ring
Tooth number	28	36	100
Tooth width/mm	10	10	10
Modulus/mm	1	1	1
Pressure angle/°	20	20	20
Mass/kg	0.0356	0.0544	0.432
Moment of inertia/(kg·m^2^)	0.00047	0.00121	0.143

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
