# Peer review of "Research on Root Strain Response Characteristics of Inner Ring of Planetary Gear Transmission System with Crack Fault"

_sensors, 2022, doi:10.3390/s23010253_

Round 1
Reviewer 1 Report
In this paper, the dynamic model of the planetary gear transmission system was established, and the failure excitation of the inner gear ring under the crack failure of the sun wheel was analyzed. Furthermore, the influence mechanism of root crack failure of the sun wheel and inner gear ring on the root strain of the inner gear ring of planetary gear transmission was also studied.
This work is important in gear fault diagnosis. Therefore, the manuscript has some merit in this work. However, there are several minor questions as follows:
1. There are many grammar wrongs in the manuscript. In addition, the language of this paper hasn't reached the international Journal requirement. Recommend that a professional native language company or a native English speaker polish the manuscript.
2. In section 1, the authors only listed references one by one and did not give any views. Recommend the authors should improve it as soon as possible.
3. In section 1, lines 82-87, the authors mentioned, "In this paper, the dynamic model of planetary gear transmission system was established,…… planetary gear transmission was studied." It can be regarded as one separate paragraph.
4. In section 2, the authors state that the finite element model of the inner gear ring was established. The authors need to provide a specific explanation. How to simulate with FEM, and which FEM commercial software was used?
5. The authors stated that the inner ring crack fault could be distinguished by fast spectrum kurtosis based on SQI company's planetary gear transmission system. It needs more evidence to prove.
6. There is no discussion section. The authors could extend the section based on section 1's citation or your proposed methods' limitations.
7. The conclusions also can be improved. The authors could give some future suggestions based on conclusions.
8. References are geographically biased, and most references are from Chinese authors.
Hopefully, this will help in the revision of the manuscript.
Reviewer 2 Report
Dear Authors,
According to the abstract, the authors undertook a very ambitious and at the same time very difficult task to determine the impact of tooth root cracks of ring and sun gears on the deformation of the ring gear rim. According to the reviewer, this is one of the reasons why the authors did not avoid several serious understatements and shortcomings in the analysis of this problem. They will be listed below in order.
1) Introduction - Several very important pieces of literature are missing, as will be shown (and proven) below.
2) Subchapter 2.1. Construction of planetary ... - The ring gear in Fig. 1(a) is wrongly and confusingly drawn as a very rigid element, because the 8 supports of this gear are not drawn and the method of support (e.g. pins or other) is not specified. This detail is very important, because only the flexible ring gear rim can be deformed under the influence of the planetary gear. Ring gear rim thickness not specified.
3) Subchapter 2.2. Construction of System Dynamics Model - The system of equations of dynamics consists of too few equations and therefore it is not possible to obtain solutions (it has 24 DOF). Planetary gears are also affected by sun gear and carrier!! It is necessary here to refer to the relevant position of the literature, and in the annex to present the derivation of these equations of dynamics.
4) Section 2.3. Solution of time-varying meshing stiffness (should rather read: Determination of time-varying meshing stiffness) is also not complete, because it does not contain the internal meshing case necessary to determine the meshing stiffness of the planetary gear - ring gear pair. In addition, when using the penalty method to analyze the contact of teeth, i.e. also to determine the stiffness of the meshing, it is required to plot a pair of contacting teeth with the FEM mesh and a description of the type of elements, e.g. Solid No., Targe No., Conta No., or others. The ring gear rim support stiffness value and possibly the method of its determination or source are also necessary. There is also no reference to the relevant literature, because it is not the authors' own method and, moreover, it has been described briefly. It was also necessary to mention other methods of determining the stiffness of the external and internal meshing. Here are some examples of literature items:
Victor Roda-Casanovaa, Ignacio Gonzalez-Perez: Investigation of the effect of contact pattern design on the mechanical and thermal behaviors of plastic-steel helical gear drives. Mechanism and Machine Theory 164 (2021) 104401
Marco Cirelli, Oliviero Giannini, Pier Paolo Valentini, Ettore Pennestri: Influence of tip relief in spur gears dynamic using multibody models with movable teeth. Mechanism and Machine Theory 152 (2020) 103948
Stephanie Seltmann, Alexander Hasse: Topology optimization of compliant mechanisms with distributed compliance (hinge-free compliant mechanisms) by using stiffness and adaptive volume constraints instead of stress constraints. Mechanism and Machine Theory 180 (2023) 105133
R.Weyler J.Oliver T.Sain J.C.Cante: On the contact domain method: A comparison of penalty and Lagrange multiplier implementations. Computer Methods in Applied Mechanics and Engineering Volumes 205–208, 15 January 2012, Pages 68-82
Zhang Xijin, Fang Zongde, Yin Xunmin, Wang Ke, and Ma Yaoguo: Research on methods of tooth contact analysis and modification optimization for internal spur gear pair. Proceedings of the Institution of Mechanical Engineers, Part C: Journal of Mechanical Engineering ScienceVolume 236, Issue 19, October 2022, Pages 10175-10184
5) Subchapter 2.4. Calculation of gear fault stiffness - Here, the authors did not mention the method of determining the stiffness at all (the use of the penalty method would be very difficult to use due to the crack). The best method for such a case is the analytical method or the use of FEM. No e value and no indication that only the case of the crack along tooth width uniformly is considered (across the entire width of the root-fillet (some cracks occur uniformly).
Xihui Liang, Ming J. Zuo, Mayank Pandey: Analytically evaluating the influence of crack on the mesh stiffness of a planetary gear set. Mechanism and Machine Theory Volume 76, June 2014, Pages 20-38
Zhanwei Lia , Hui Maa, Mengjiao Fenga , Yunpeng Zhuc , Bangchun Wen . Meshing characteristics of spur gear pair under different crack types.. Engineering Failure Analysis Volume 80, October 2017, Pages 123-140
6) Dynamic load analysis ... Poprawnośc wykresów tego rozdziału bedzie mozliwa do sprawdzenia dopiero po poprawie (uzupełnieniu) układu równań dynamiki.
7) Chapter 4. Solution of response of finite element model of inner ring and Chapter 6. Strain analysis - The load determined from the dynamic model was used to determine the deformation of the ring gear rim. An incomplete system of equations does not ensure the correctness of the solution. A little note: if the authors used to call pinion sun gear, why do they now call it solar gear (a less common term). After the system of dynamics equations has been corrected, the literature on determining the deformation of the flexible ring gear rim should be added, because it exists.
Zaigang Chen, Yimin Shao: Mesh stiffness of an internal spur gear pair with ring gear rim deformation. Mechanism and Machine Theory Volume 69, November 2013, Pages 1-12
8) Chapter 5. Experimental verification was carried out independently of the calculations, so it should be correct.
Conclusion: The chapters on the dynamic model and stiffness determination should be thoroughly revised.
Sincerely, Reviewer.

Reviewer 3 Report
The article has a scientific character. The article deals with the vibroacoustic properties of a planetary gear with tooth craks on the central gear. The Authors applied correct research methods and used the appropriate measuring equipment. The content of the work is logically written. The manuscript contains 25 figures and 1 table. Figures and tables are properly prepared. Authors cited 22 literature sources. The authors presented an interesting work, but it requires numerous improvements to be of satisfactory quality.
General remarks
1. The authors must clearly indicate the novelty of the work
2. The work also requires careful text editing
3. The authors must explain how the stiffness determined for wheels with external gearing corresponds to the stiffness of wheels with internal gearing, which were tested in the paper.
4. The authors must explain why they did not take into account the influence of the sun gear on the dynamics of the gear; in systems with four satellites it is more difficult to obtain the optimal distribution of satellites cooperating with the sun gear (this is clearly visible in Fig. 14)
5. The authors must provide how the dynamic model of the planetary gear was determined, the work does not specify the source and the method of its determination.
6. The stiffness of the meshing is usually determined differently than presented by the authors, in particular the stiffness depends on the meshing index, and this issue was not taken into account by the authors (only in Fig. 4 zones of one and two pairs of meshing are drawn).
Detailed comments
Row 90 The authors must state in what system the tested wind turbine gearboxes work: as a multiplier or as a reducer?
Figure 1b - in the diagram Authors must show where is the entrance and where is the exit?
Table 1 should give the gear ratio and gear ratio and frontal mesh ratio epsilon
Figure 3 - how the model relates to the tested system of planetary gears - inner ring
Point 2.4 should be given how the results presented in Fig. 6 were obtained (no reference to this figure in the text)
In order to publish the paper, the Authors must correct the indicated reservations and explain the issues raised in the text in general remarks
Round 2
Reviewer 1 Report
The authors have responded most of the points.
Reviewer 3 Report
Dear Authors,
Thank you for considering my comments. I will recommend publishing the work.